# A Comparative Study on the Distribution Pattern of Endocrine Cells in the Gastrointestinal Tract of Two Small Alpine Mammals, Plateau Zokor (*Eospalax baileyi*) and Plateau Pika (*Ochotona curzoniae*)

**DOI:** 10.3390/ani13040640

**Published:** 2023-02-12

**Authors:** Xincheng Cai, Darhan Bao, Rui Hua, Bin Cai, Lei Wang, Rui Dong, Limin Hua

**Affiliations:** College of Grassland Science, Gansu Agricultural University, Key Laboratory of Grassland Ecosystem of the Ministry of Education, Engineering and Technology Research Center for Alpine Rodent Pests Control, National Forestry and Grassland Administration, Lanzhou 730070, China

**Keywords:** plateau zokor, plateau pika, gastrointestinal tract, argyrophilic cells, immunoreactive cells, adaptability

## Abstract

**Simple Summary:**

The gastrointestinal tract (GIT) is the largest and most complex endocrine organ; there are numerous endocrine cells in the mucosal layer of gastrointestinal tissue, which can secrete a variety of gastrointestinal hormones that regulate gastrointestinal digestion and absorption. The small alpine mammals, plateau zokor (*Eospalax baileyi*) and plateau pika (*Ochotona curzoniae*), live in the unique ecotope with cold, hypoxic environments and short plant-growing seasons, resulting in their differential adaptive digestive strategies for foods. In this study, we compared the distribution pattern of argyrophilic cells and the expression of 5-HT cells, Gas cells, and Glu cells in the GIT of the plateau zokor and plateau pika by using histochemical and immunohistochemical methods. Our results showed that the distribution pattern of endocrine cells in the GIT of plateau zokor and plateau pika not only have specific interspecies differences, but also exhibit parallel adaptation to the alpine environments. In addition, we conclude that the distribution pattern of endocrine cells in the GIT is consistent with the respective animals’ diets.

**Abstract:**

Endocrine cells can secrete a variety of gastrointestinal hormones that regulate gastrointestinal digestion and absorption, which, in turn, play an important role in animal growth, metabolism, and acclimation. The small alpine mammals, plateau zokor (*Eospalax baileyi*) and plateau pika (*Ochotona curzoniae*), live in a unique ecotope with cold, hypoxic environments and short plant-growing seasons, resulting in differential adaptive digestive strategies for foods. Studying the distribution pattern of endocrine cells in the gastrointestinal tract (GIT) of these two animals can lead to a better understanding of the survival strategies of animals in an alpine environment. In this study, we used histochemical and immunohistochemical methods to compare the distribution pattern of argyrophilic cells and the expression of 5-HT cells, Gas cells, and Glu cells in the GIT of the plateau zokor with those of the plateau pika. The results showed that these endocrine cells we studied were widely distributed in the gastrointestinal organs of both these small mammals, and their morphology and distribution location in the GIT were almost the same. However, there were significant differences in the distribution density of argyrophilic cells between different organs in the GIT. The distribution density of argyrophilic cells in the duodenum, jejunum, ileum, and rectum of plateau zokor was significantly lower than that of plateau pika (*p* < 0.05) and, in the cecum of plateau zokor, was significantly higher than that of plateau pika (*p* < 0.001). The positive expression of 5-HT cells in the corpus I, corpus II, and pylorus of the stomach, duodenum, ileum, and rectum of plateau zokor was significantly higher than that of plateau pika (*p* < 0.01). In addition, the positive expression of Glu cells in the cecum was significantly higher (*p* < 0.01) and in the duodenum and colon was significantly lower (*p* < 0.05) in the plateau zokor than in the plateau pika. We conclude that the distribution pattern of endocrine cells in the GIT is consistent with the respective animals’ diets, with the plateau zokor feeding on high-fiber roots and plateau pika preferring to intake the aboveground parts of plants with lower fibers.

## 1. Introduction

The gastrointestinal tract (GIT) is not only the main component of the digestive system in animal species, but also the largest and most complex endocrine organ [1,2]. There are a large number of endocrine cells in the mucosal layer of gastrointestinal tissue. These endocrine cells can secrete a variety of gastrointestinal hormones that regulate gastrointestinal digestion and absorption, which play an important role in animal growth, metabolism, and adaptation to external environments [3,4]. In recent decades, scholars have conducted extensive research on the identification, function, and adaptability of endocrine cells in the digestive tract of avians, fishes, amphibians, and mammals [5,6,7,8]. It is generally agreed that the regional distribution, relative density, and cell types of endocrine cells in the digestive tract are closely related to the habitat, foods, and lifestyle of animals, and there are obvious specific interspecies differences [9,10,11].

Argyrophilic cells are a population of endocrine cells and are widely distributed in the GIT of vertebrates [12]. The gastrointestinal hormones secreted by argyrophilic cells perform specific physiological functions, such as the protection of gastrointestinal mucosa and the regulation of digestive function [13]. The morphology of argyrophilic cells is closely related to their function. Round-shaped and oval-shaped cells are called closed-type cells because they have no direct contact with the gastrointestinal lumen, which performs endocrine functions. The other types of cells are called opened-type cells and they perform exocrine functions. These opened-type cells are distributed throughout the GIT, mainly in the form of cone-shaped and spindle-shaped cells, and the top of such cells can extend into the gastrointestinal lumen to sense stimuli in the internal environments of the GIT [14]. Many studies have reported the distribution pattern of argentophilic cells in the GIT of animals, and they have generally found that mammalian argyrophilic cells have both endocrine and exocrine functions [15]. Additionally, the differences have been found in the distribution density of argentophilic cells in various organs of the GIT, with the overall pattern being stomach > small intestine > large intestine [16].

Endocrine cells, the major sensors of luminal contents in the GIT, secrete various hormones, including 5-hydroxytryptamine (5-HT), gastrin (Gas), and glucagon (Glu), that regulate a wide range of biological and physiological processes locally or remotely [17]. The hormones 5-HT, Gas, and Glu are commonly found in the GIT of mammals and are involved in appetite, food ingestion and digestion, gastrointestinal motility, metabolism, and immune responses [18,19]. Within them, 5-HT can regulate smooth muscle contraction, mucus secretion and vasodilation in the GIT [20], whereas Gas can stimulate the secretion of pepsin and gastric acid, and simultaneously promote the growth and differentiation of the mucosal epithelium in the gastrointestinal tissues [21]. In addition, Glu can enhance the growth of gastrointestinal mucosa, repair damaged mucosa, and inhibits gastrointestinal motility and gastric acid secretion to protect the mucosal layer of the GIT effectively [19]. To date, the endocrine cells and the gastrointestinal hormones secreted by them in the GIT of mammalians have been extensively studied by scholars, and they agreed that the expression of these hormones exhibited local and regional variations along the GIT of animals, with highlighted substantial heterogeneity [8].

As a unique subterranean rodent, the plateau zokor (*Eospalax baileyi*) spends most of its life underground in closed tunnel systems and feeds primarily on forb rhizomes [22]. The plateau pika (*Ochotona curzoniae*) is a small fossorial lagomorph that forages mostly on fresh stems, leaves, and buds of aboveground plants [23]. The plateau zokor and plateau pika have a sympatric distribution and are the two dominant herbivorous rodents living in the alpine meadow of the Qinghai–Tibet Plateau, China. They are often used as the ideal animal model to study the adaptability of animals to low oxygen and cold environments because they have formed a series of special mechanisms to adapt to the alpine environments through a long-term evolutionary process [24]. To date, researchers have conducted a large number of in-depth and meaningful studies on the adaptability of the plateau zokors and plateau pikas to the plateau environments in terms of morphology, physiology, and behavior [25,26]. However, the distribution pattern of endocrine cells in these two species and how they adapt to the different foods available in their environments have not been reported thus far.

In this study, we compared the distribution characteristics of argyrophilic cells and the expression of 5-HT-immunoreactive cells, Gas-immunoreactive cells, and Glu-immunoreactive cells in the GIT of the plateau zokor and plateau pika by using histochemical and immunohistochemical methods. The purpose of this study was to preliminarily explore the adaptation strategies of endocrine cells in the GIT of these two animals to their different foods and to provide a reference on the physiological digestive mechanism of small mammals living in alpine environments.

## 2. Materials and Methods

### 2.1. Experimental Animals and Their Habitats

Seven adult plateau zokors (three males and four females; the average weight was 351.80 ± 19.00) and six plateau pikas (three males and three females; the average weight was 177.15 ± 6.11) were captured by traps during their nonbreeding season in October 2020 in Maqu County (Gansu Province, China), which is located in the eastern Qinghai–Tibet Plateau. The sampling area was an alpine meadow with an altitude of 3434 m (Figure 1A). There is a large temperature difference between night and day and the climate is cold and humid. For the alpine climate, there are only two seasons (cold and warm) throughout the year, and the plant-growing period sustains approximately 190 days [27]. In this area, there are two indigenous small mammals with sympatric distributions: the plateau zokor (Figure 1A) and plateau pika (Figure 1B). However, the plateau zokor feeds perennially on the rhizomes of forbs [28] and the plateau pika prefers to feed on fresh stems and leafy parts of Poaceae and Cyperaceae [29].

### 2.2. Tissue Preparation

The GIT was dissected after the experimental animals were euthanized. Afterward, each organ (the fundus, corpus I, corpus II, and pylorus of the stomach, duodenum, jejunum, ileum, colon, rectum, and cecum) of the GIT was excised into small tissue fragments (5 mm) [25]. These tissues were then immediately fixed in 4% paraformaldehyde for at least 24 h. The fixed gastrointestinal tissue fragments were embedded in paraffin wax and sliced into 6 μm thick sections with a microtome (Leica-2016, Leica, Wetzlar, Germany).

The protocols and procedures for animal experiments in the study were approved by the Institutional Animal Care and Usage Committees of the Grassland Science College of Gansu Agricultural University (GSC-IACUC-2020-0015).

### 2.3. Histochemical Study

For the histochemical study, the previously prepared paraffin sections were subjected to gradient dewaxing and rehydration, followed by incubation with silver-solution incubation medium (taking 3 mL 1% silver nitrate + 10 mL acetate buffer + 87 mL ultrapure water. 1% silver nitrate solution: 1 g silver nitrate + 100 mL ultrapure water; acetate buffer: solution A (1.21 mL glacial acetic acid + 98.7 mL ultrapure water) mixed with solution B (2.72 g sodium acetate + 100 mL ultrapure water) = 1:9) at 60 °C for 3 h, and then reduction with a reducing solution (1 g hydroquinone + 2.5 g anhydrous sodium sulfite + 100 mL ultra-pure water) at 45 °C for 1 h [13,30]. Finally, dehydration, transparency, and sealing with neutral gum were performed. Microphotographs of each section were obtained using a digital section scanner (Pannoramic 250, Danjier, Jinan, China) for the histochemical analysis. Ten microphotographs (400× magnification) were randomly selected from each part of each sample to observe the distribution characteristics of argyrophilic cells in tissue sections.

### 2.4. Immunohistochemical Study

For the immunohistochemical study, the tissue sections after dewaxing and rehydration were heated for 15 min in citrate buffer (0.01 mol/L, pH = 6.0) to recover the antigens. Subsequently, the tissue sections were placed in 3% H_2_O_2_ for 15 min to incubate, and then washed with phosphate-buffered saline (PBS, 0.01 mol/L). Afterward, the tissue sections were incubated in goat serum blocking solution at room temperature for 30 min, and then incubated overnight at 4 °C with the primary antisera (anti-serotonin, anti-glucagon, and anti-gastrin; dilution ratio = 1:200). Then, the sections were incubated with a biotinylated secondary antibody, followed by staining with DAB colorimetric solution (dilution ratio = 1:20) at room temperature and re-staining with hematoxylin staining solution [18,31]. Finally, the sections were subjected to dehydration, immersed in a clearing agent to enhance their transparency, and sealed with neutral gum. All of the above reagents were from Beijing Zhongshan Golden Bridge Biological Company, China. Three microphotographs (400× magnification) were randomly acquired from each part of each sample by using a Digital trinocular microscope camera system (BA400Digital, Motic, Xiamen, China). The distribution characteristics of 5-HT-immunoreactive cells, Gas-immunoreactive cells, and Glu-immunoreactive cells were observed and their positive expression levels (the proportion of DAB positive tissue = DAB positive area/tissue area × 100%) were calculated using the Data Image Analysis System (Halo 101-WL-HALO-1, Indica labs, Albuquerque, NM, USA).

### 2.5. Data Analysis

All figures were completed with GraphPad prism 8.0 after data were preprocessed in Excel 2016 and analyzed in SPSS 19.0 software. Before statistical analysis, we used the Kolmogorov–Smirnov test and Levene test to determine a normal distribution and homogeneity of variance of the original data. A comparison of the interspecific differences between the plateau zokor and the plateau pika, in terms of the quantitative characteristics of endocrine cells in the GIT, was performed with an independent samples *t*-test (*a* = 0.05). To evaluate intraspecific differences in the quantitative characteristics of endocrine cells in the gastrointestinal organs of the plateau zokor and the plateau pika, multiple comparisons were made using the least significant difference (LSD) method (*a* = 0.05). The data in this paper are expressed as mean ± standard error (Mean ± SE), with *p* < 0.05 indicating that the difference is significant. In this paper, the data are represented by mean ± standard error (Mean ± SE).

## 3. Results

### 3.1. Distribution Pattern of Gastrointestinal Argyrophilic Cells

After silver staining, the gastrointestinal argyrophilic cells of the plateau zokor and plateau pika were brown under the light microscope. The argyrophilic cells existed throughout acinar cells and epithelial cells in the tissue mucosa and were widely distributed in all regions of the GIT of the plateau zokor and plateau pika. Two types of argyrophilic cells were observed: closed-type argyrophilic cells, with a morphology that was mainly round-shaped or oval-shaped, and the opened-type argyrophilic cells, with a mainly cone-shaped or spindle-shaped morphology (Figure 2). Comparison of the interspecific differences in the gastrointestinal organs showed that the proportion of opened-type argyrophilic cells in the colon and rectum of the plateau zokor was significantly lower than that in the corresponding organs of the plateau pika (*p* < 0.05, Figure 3B), while there was no significant difference in other organs between the two animals (Figure 3).

There were obvious interspecific differences between the plateau zokor and plateau pika in the distribution density of argyrophilic cells in gastrointestinal organs (Table 1). In the stomach, argyrophilic cells were only present in the corpus II of the stomach region in the plateau zokor, and the density was significantly more than the corresponding regions in the plateau pika (*p* < 0.05), while argyrophilic cells were present throughout the whole stomach in the plateau pika. In addition, the argyrophilic cells were present throughout the entire intestinal tract of the plateau zokor and plateau pika but with some interspecific differences in the distribution density of cells. The density of argyrophilic cells in the duodenum, jejunum, ileum, and rectum of the plateau zokor was significantly lower than that in the plateau pika (*p* < 0.05), but the density of argyrophilic cells in the cecum was significantly higher than that in the plateau pika (*p* < 0.001). There was no significant difference in the density of argyrophilic cells in the colon between the two animals.

Intraspecific differences in the distribution density of argyrophilic cells in the GIT of the plateau zokor and plateau pika were also apparent (Table 1). In the GIT of the plateau zokor, the density of argyrophilic cells was highest in the corpus II of the stomach, followed by the colon, and was lowest in the rectum, with no significant differences between the duodenum, jejunum, ileum, and cecum. However, in the plateau pika, the density of argyrophilic cells was the highest in the fundus of the stomach, and the cecum and rectum were low, which was significantly different from the density in the other gastrointestinal organs (*p* < 0.05).

### 3.2. Differences in the Positive Expression of Gastrointestinal Immunoreactive Cells

The 5-HT-immunoreactive cells, Gas-immunoreactive cells, and Glu-immunoreactive cells were brown after staining and widely distributed throughout the mucosal epithelium of various regions of the GIT in both the plateau zokor and plateau pika. The morphology of these three types of endocrine cells was mainly cone-shaped and spindle-shaped, which belong to opened-type cells (Figure 4, Figure 5, and Figure 6).

### 3.2.1. 5-HT-Immunoreactive Cells

The positive expression levels of 5-HT-immunoreactive cells in all organs of the GIT of the plateau zokor were higher than those in the corresponding regions of the plateau pika, except for in the cecum, and those in the corpus I of the stomach, corpus II of the stomach, and pylorus of the stomach, duodenum, ileum, and rectum were significantly higher than those of the plateau pika (*p* < 0.05, Figure 7). 

#### 3.2.2. Gas-Immunoreactive Cells

The positive expression levels of Gas-immunoreactive cells in the stomach, duodenum, and jejunum of the plateau zokor were higher than those of the plateau pika, while those in the ileum, colon, rectum, and cecum were lower in the plateau zokor than in the plateau pika, but there was no significant difference between the two species (Figure 8).

#### 3.2.3. Glu-Immunoreactive Cells

For Glu-immunoreactive cells, the positive expression levels in the corpus I and corpus II of the stomach were higher in the plateau zokor than in the plateau pika, while those in the fundus and pylorus of the stomach were lower in the plateau zokor than in the plateau pika, but there was no significant interspecific difference (Figure 9). In the intestinal tract, the positive expression levels of Glu-immunoreactive cells in the cecum were significantly higher in the plateau zokor than in the plateau pika (*p* < 0.01), while those in the duodenum and colon were significantly lower in the plateau pika than in the plateau pika (*p* < 0.05), and there was no significant interspecific difference in other intestinal organs.

## 4. Discussion

The study of the distribution of endocrine cells is the foundation for understanding the physiological regulation mechanism of digestion in the animal GIT. It is also an important approach for elucidating the strategies exhibited by animals in terms of the adaptation of their gastrointestinal functions to external environments. Previous studies have shown that the endocrine cells can regulate various physiological processes, such as nutrient ingestion, metabolism, and gastrointestinal motility by secreting more than 20 types of gastrointestinal hormones [32,33]. Studies have also shown that the regional distribution and density of endocrine cells in the GIT are affected by the habitat, diets, and lifestyle of animals [6,9]. In this study, we compared the distribution pattern of argyrophilic cells, 5-HT cells, Gas cells, and Glu cells in the GIT of the plateau zokor and plateau pika. The results showed that the endocrine cells we detected were widely distributed in the GIT of these two species, and their location and morphology of distribution were broadly similar. However, there were obvious interspecific differences in the distribution density of endocrine cells in each gastrointestinal organ.

The argyrophilic cells, as an important type of endocrine cell, are commonly found in the GIT of mammals [34], and this is no exception but rather more critical in the case of the plateau zokor and plateau pika, which are small mammals adapted to the alpine environments. The results of this study showed that argyrophilic cells were distributed throughout the whole intestinal tract of the plateau zokor and plateau pika, but there were differences in the stomach. Among these differences, the argyrophilic cells were only found in the corpus II of the stomach in the plateau zokor, while the argyrophilic cells were detected in all four parts of the plateau pika’s stomach. This may be due to significant differences in the morphology and histology of the stomach between the two animals. During adaptation to its high-fiber diets, the plateau zokor has evolved a double-chambered and hemi-glandular stomach, in which only the corpus II of the stomach has a thicker mucosal layer, while the other parts of the stomach have a thicker muscular layer and a very thin mucosal layer [25]. The argyrophilic cells mainly exist throughout the epithelial cells and acinar cells of the mucosal layer and regulate gastrointestinal function by secreting hormones [14]. The thin mucosa of the fundus, corpus I, and pylorus of the stomach cannot provide attachment conditions for the argyrophilic cells. Furthermore, these sites mainly cause gastric motility through the contraction of the thicker muscle to perform the preliminary mechanical digestion of foods, and almost no argyrophilic cells secrete hormones to stimulate the stomach for chemical regulation. 

There were also intraspecific differences in the distribution density of argyrophilic cells in the various gastrointestinal organs of the plateau zokor and plateau pika. The results showed that the density of argyrophilic cells in the GIT of the plateau zokor and plateau pika was greatest in the stomach in both species, but whereas, in the plateau zokor, it was greatest in the corpus II of the stomach, in the plateau pika, it was greatest in the fundus of the stomach. The distribution density of argyrophilic cells was lowest in the rectum of the plateau zokor, while, in the plateau pika, the density was lowest in the rectum and cecum. The distribution density of argyrophilic cells in various regions of the GIT is related to the difference in the food composition and digestion pattern of animals [35]. The food quality of the plateau zokor is usually poor and it is especially rich in fibers. The high-fiber foods are first mechanically processed in the fore-stomach (mainly located in the fundus of stomach), and then the food mass enters the hind-stomach (mainly located in the corpus II of stomach) for chemical digestion, with the plentiful argyrophilic cells secreting gastrointestinal hormones to stimulate gastric acid and mucus. However, the plateau pika’s foods are relatively easy to digest, and the mechanical digestion and chemical digestion of the food are carried out simultaneously in the stomach. The gastric fundus has the thickest mucosa, and this location is the main region where the numerous argyrophilic cells contribute to the regulation of digestion. As for the lowest distribution density of argyrophilic cells being in the rectum of the two animals, again, this is related to the function of the organ; because the rectum is an organ for storing food residues and excreting waste, the digestion and absorption function is extremely weak [36], so there are few argyrophilic cells. Moreover, the cecum digests indigestible components, such as cellulose and lignin, mainly through microbial fermentation [37], and the distribution density of argyrophilic cells in the cecum of the plateau pika was consistent with its digestive function.

There are different response strategies for various gastrointestinal organs to individual diets in the process of digestion and absorption [38]. Many studies have shown that the stomach and cecum are greatly affected by the quality of the diets [39]. The plateau zokors live in underground tunnels for a long time; compared with plateau pikas, they occupy a poor niche for food resources [40]. Our results showed that the density of argyrophilic cells in the corpus II of the stomach and the cecum was significantly higher in the plateau zokor than in the plateau pika. This suggests that the presence of more argyrophilic cells in these organs is a beneficial adaptation to its high-fiber diets. Furthermore, although the plateau zokor and plateau pika, two herbivorous rodents, both live in a cold and hypoxic environment, they face different energy pressures due to their different respective lifestyles [41]. Compared with the plateau zokor, the plateau pika is a small mammal with a high metabolic and gastrointestinal-emptying rate due to its extremely energy-consuming activities, such as daily foraging, avoiding natural enemies, and maintaining body temperature [42]. As the small intestine is the main place for nutrient absorption, the higher distribution density of argyrophilic cells in it can improve the efficiency with which the plateau pika absorbs nutrients and, hence, enhance its energy supplement. Similarly, the rectum is an organ for excreting waste, and the higher density of argyrophilic cells here can accelerate intestinal emptying in the plateau pika. Finally, we also found that the proportion of opened-type argyrophilic cells in the colon and rectum was higher in the plateau pika than in the plateau zokor. The opened-type argyrophilic cells are in direct contact with the intestinal lumen through the prolongation of microvilli, and these cells release gut hormones via the activation of the luminal to perform the digestive function [4]. The results of a higher proportion of opened-type argyrophilic cells may also be the functional adaptation of the argyrophilic cells in the GIT to hypermetabolism of plateau pika.

The GIT is the most important region for 5-HT, Gas, and Glu activities, and these three hormones have been shown to have a strong effect on gastrointestinal regulation and digestive function [43]. In this study, the 5-HT-immunoreactive cells, Gas-immunoreactive cells, and Glu-immunoreactive cells were widely distributed in the gastrointestinal organs of the plateau zokors and plateau pikas, but there were obvious interspecific differences in the positive expression of these cells. The positive expression of 5-HT-immunoreactive cells in the gastrointestinal organs was generally higher in the plateau zokor than that in the plateau pika. The diet of the plateau zokor is rich in cellulose and lignin, and having more 5-HT-immunoreactive cells will promote gastrointestinal motility and mucus production by increasing the secretion of hormones, which not only ensures the digestion of rough foods, but also ensures the smooth removal of waste residues. In addition, Zhang and Chen [44] found that aerobic exercise can reduce the density of endocrine cells in the gastrointestinal tract of rats (*Rattus norvegicus*). Compared with the plateau zokor, a subterranean rodent with limited activity, the higher activity intensity of the plateau pika will induce gastrointestinal motility and, thus, require fewer 5-HT-immunoreactive cells. 

Although there was no significant interspecific difference in the expression of Gas-immunoreactive cells between the GIT of the plateau zokor and plateau pika, our results showed that the expression of Gas-immunoreactive cells in the foregut was higher in the plateau zokor than in the plateau pika, while the expression in the hindgut was the opposite. This may be the compensatory adaptation of the endocrine cells in various organs of the GIT to the habitat, diets, and lifestyle of these two animals [9,10,44]. Indications of adaptation were also apparent in the expression of Glu-immunoreactive cells. The positive expression of these cells in the small intestine was significantly lower in the plateau zokor than in the plateau pika, while that in the cecum was significantly higher in the plateau zokor than in the plateau pika. The hormones secreted by Glu-immunoreactive cells can enhance the growth and repair of the gastrointestinal mucosa [19]; on the other hand, they can stimulate glycogen decomposition to increase blood glucose levels [45]. The distribution characteristics of Glu-immunoreactive cells in the small intestine may, therefore, be related to animal metabolism, and that in the cecum may be related to food composition. We hypothesized that these results may indicate the species-dependent variation of Glu-immunoreactive cells in the GIT in the two animals related to their diets and lifestyles, but further verification is needed.

## 5. Conclusions

In summary, our results show that these endocrine cells are widely distributed in the gastrointestinal organs of the plateau zokor and plateau pika, and their morphology and location are broadly similar, but there are compensatory differences in the distribution density of each gastrointestinal organ. Based on the results of this study, we believe that the distribution pattern of endocrine cells in the GIT of plateau zokor and plateau pika not only have specific interspecies differences, but also exhibit parallel adaptation to the alpine environments. In addition, we conclude that the distribution pattern of endocrine cells in the GIT is consistent with the respective animals’ diets. However, our study only described the distribution characteristics of a limited range of endocrine cells in the GIT of the plateau zokor and plateau pika and included only a preliminary exploration of their adaptive digestion strategies. The physiological mechanism by which endocrine cells regulate gastrointestinal digestion in mammals living in alpine environments deserves further study.

## Figures and Tables

**Figure 1 animals-13-00640-f001:**
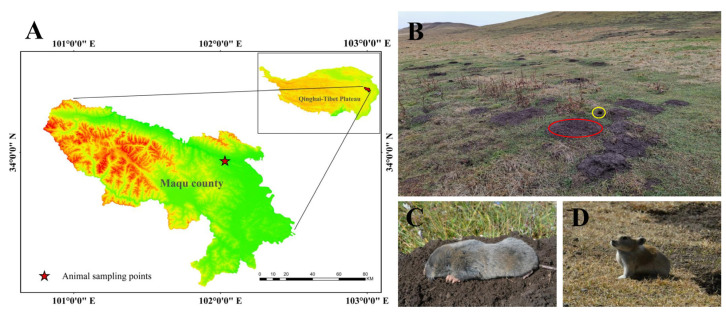
Experimental animals and their habitats in this study. (**A**) The sampling location of experimental animals; (**B**) the habitat of sympatric distribution of the plateau zokor and plateau pika (the red circle indicates a mound made by the plateau zokor and the yellow circle indicates a hole made by the plateau pika); (**C**) plateau zokor; (**D**) plateau pika.

**Figure 2 animals-13-00640-f002:**
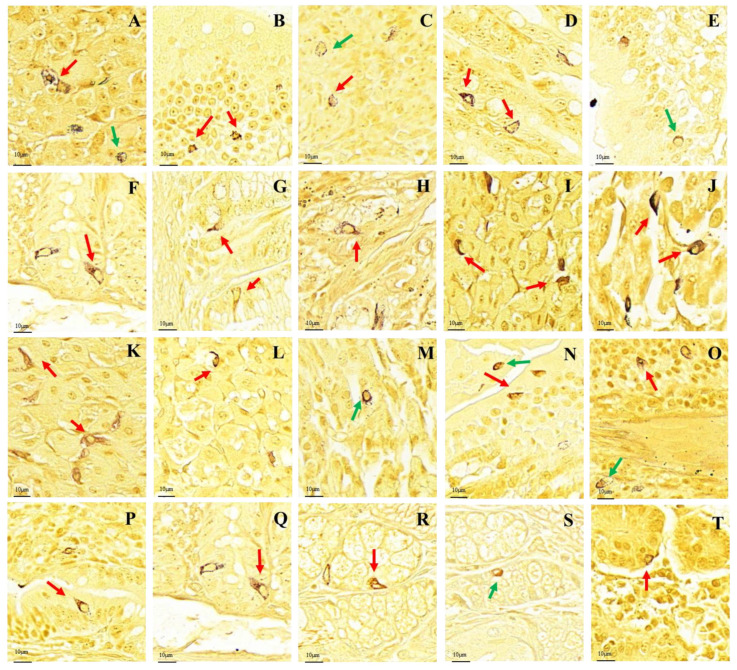
Distribution and morphology of argyrophilic cells in the GIT of the plateau zokor and plateau pika (400× magnification). (**A**–**H**) Plateau zokor ((**A**) round-shaped and cone-shaped argyrophilic cells in the corpus II of stomach; (**B**) cone-shaped and spindle-shaped argyrophilic cells in the duodenum; (**C**) oval-shaped and cone-shaped argyrophilic cells in the jejunum; (**D**) cone-shaped argyrophilic cells in the ileum; (**E**) round-shaped argyrophilic cells in the ileum; (**F**) cone-shaped argyrophilic cells in the colon; (**G**) cone-shaped and spindle-shaped argyrophilic cells in the rectum; (**H**) cone-shaped argyrophilic cells in the cecum); (**I**–**T**) plateau pika ((**I**) cone-shaped and spindle-shaped argyrophilic cells in the fundus of stomach; (**J**) cone-shaped and spindle-shaped argyrophilic cells in the corpus I of stomach; (**K**) cone-shaped and spindle-shaped argyrophilic cells in the corpus II of stomach; (**L**) cone-shaped argyrophilic cells in the pylorus of stomach; (**M**) round-shaped argyrophilic cells in the pylorus of stomach; (**N**) oval-shape and cone-shaped argyrophilic cells in the duodenum; (**O**) oval-shape and cone-shaped argyrophilic cells in the jejunum; (**P**) cone-shaped argyrophilic cells in the ileum; (**Q**) cone-shaped argyrophilic cells in the colon; (**R**) cone-shaped argyrophilic cells in the rectum; (**S**) oval-shaped argyrophilic cells in the rectum; (**T**) cone-shaped argyrophilic cells in the cecum). The red arrow indicates opened-type argyrophilic cells and the green arrow indicates closed-type argyrophilic cells.

**Figure 3 animals-13-00640-f003:**
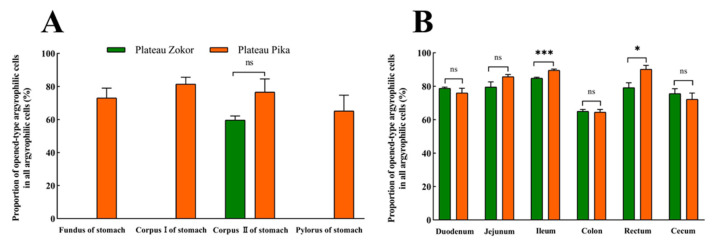
The difference in the proportion of opened-type argyrophilic cells in the gastrointestinal organs of the plateau zokor and plateau pika. (**A**) The difference in the proportion of opened-type argyrophilic cells in the stomach; (**B**) the difference in the proportion of opened-type argyrophilic cells in the intestinal tract. * and *** indicate significant interspecific difference (*p* < 0.05, *p* < 0.001); ns indicates an insignificant interspecific difference.

**Figure 4 animals-13-00640-f004:**
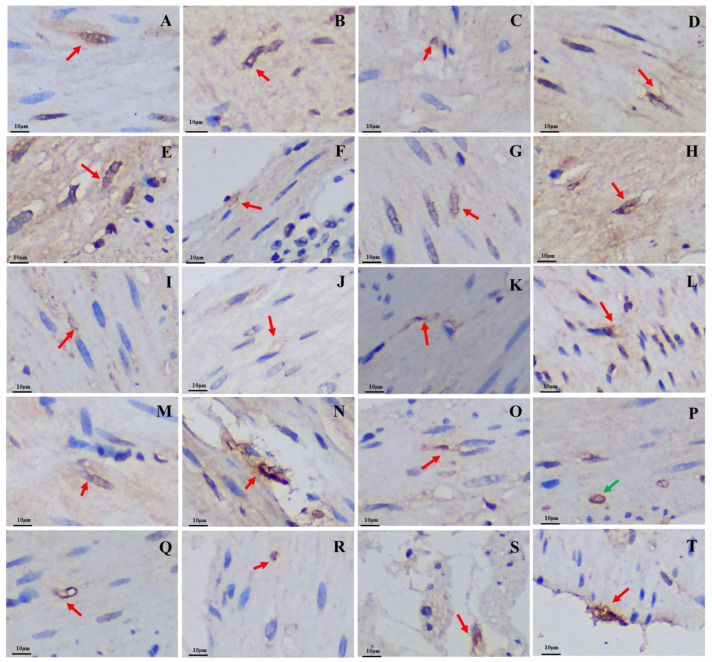
Distribution and morphology of 5-HT-immunoreactive cells in the GIT of the plateau zokor and plateau pika (400× magnification). (**A**–**J**) Followed by the fundus of stomach, corpus I of stomach, corpus II of stomach, pylorus of stomach, duodenum, jejunum, ileum, colon, rectum, and cecum of the plateau zokor; (**K**–**T**) similarly, the plateau pika’s gastrointestinal organs. The red arrow indicates opened-type argyrophilic cells and the green arrow indicates closed-type argyrophilic cells.

**Figure 5 animals-13-00640-f005:**
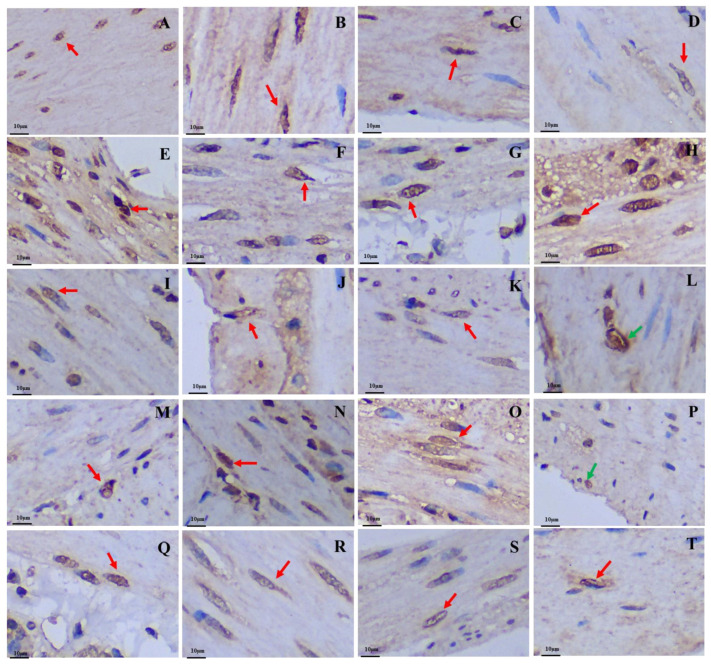
Distribution and morphology of Gas-immunoreactive cells in the GIT of the plateau zokor and plateau pika (400× magnification). (**A**–**J**) Followed by the fundus of stomach, corpus I of stomach, corpus II of stomach, pylorus of stomach, duodenum, jejunum, ileum, colon, rectum, and cecum of the plateau zokor; (**K**–**T**) similarly, the plateau pika’s gastrointestinal organs. The red arrow indicates opened-type argyrophilic cells and the green arrow indicates closed-type argyrophilic cells.

**Figure 6 animals-13-00640-f006:**
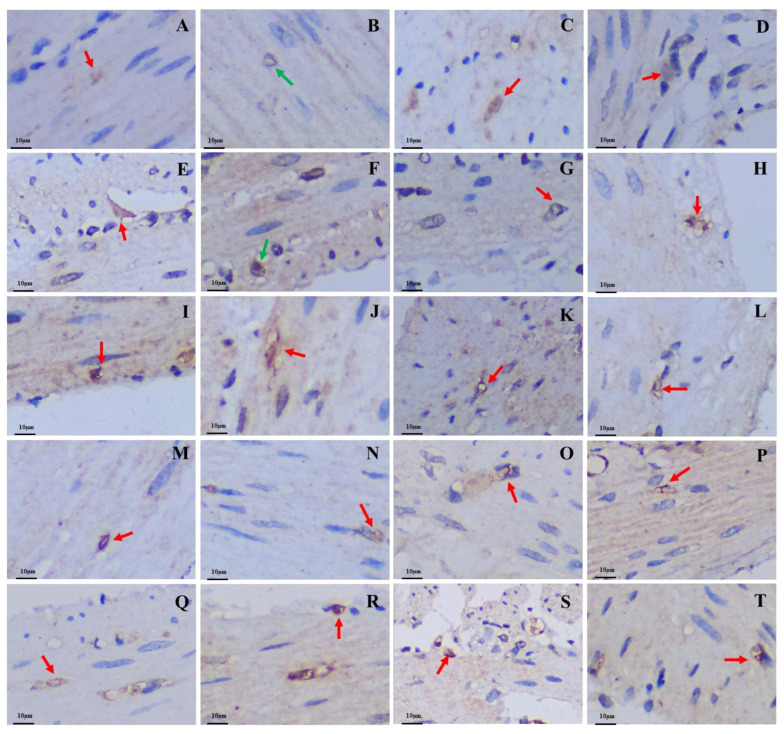
Distribution and morphology of Glu-immunoreactive cells in the GIT of the plateau zokor and plateau pika (400× magnification). (**A**–**J**) Followed by the fundus of stomach, corpus I of stomach, corpus II of stomach, pylorus of stomach, duodenum, jejunum, ileum, colon, rectum, and cecum of the plateau zokor; (**K**–**T**) similarly, the plateau pika’s gastrointestinal organs. The red arrow indicates opened-type argyrophilic cells and the green arrow indicates closed-type argyrophilic cells.

**Figure 7 animals-13-00640-f007:**
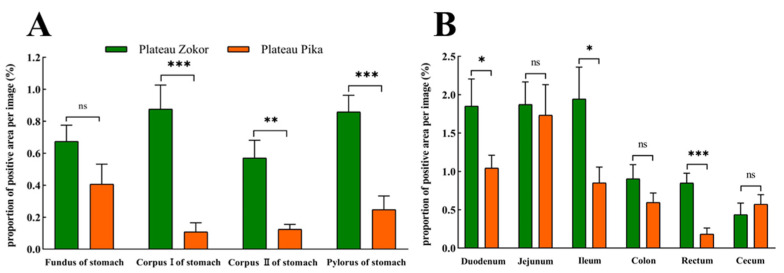
The difference in positive expression levels for 5-HT-immunoreactive cells in the gastrointestinal organs of the plateau zokor and plateau pika. (**A**) In the stomach; (**B**) in the intestinal tract. *, **, and *** indicate significant interspecific difference (*p* < 0.05, *p* < 0.01, *p* < 0.001); ns indicates an insignificant interspecific difference.

**Figure 8 animals-13-00640-f008:**
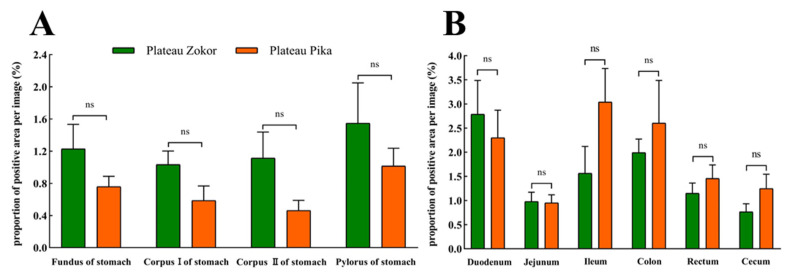
The difference in positive expression levels for Gas-immunoreactive cells in the gastrointestinal organs of the plateau zokor and plateau pika. (**A**) In the stomach; (**B**) in the intestinal tract. ns indicates an insignificant interspecific difference.

**Figure 9 animals-13-00640-f009:**
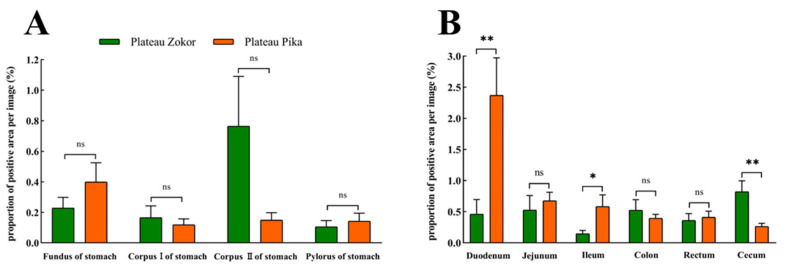
The difference in positive expression levels for Glu-immunoreactive cells in the gastrointestinal organs of the plateau zokor and plateau pika. (**A**) In the stomach; (**B**) in the intestinal tract. * and ** indicate significant interspecific difference (*p* < 0.05, *p* < 0.01); ns indicates an insignificant interspecific difference.

**Table 1 animals-13-00640-t001:** The distribution density of argyrophilic cells in GIT of plateau zokor and plateau pika (individuals/view).

Location	Plateau Zokor (*Eospalax baileyi*)	Plateau Pika (*Ochotona curzoniae*)	*t* Value	Significance
Fundus of stomach	-	32.58 ± 2.48 a	-	-
Corpus I of stomach	-	17.28 ± 3.72 b	-	-
Corpus II of stomach	33.70 ± 2.23 a	22.30 ± 5.25 b	−2.23	*
Pylorus of stomach	-	8.60 ± 1.41 c	-	-
Duodenum	8.07 ± 0.34 b	15.72 ± 1.40 b	5.32	**
Jejunum	7.47 ± 0.88 b	18.89 ± 2.34 b	4.57	**
Ileum	9.43 ± 0.16 b	16.35 ± 1.51 b	4.54	*
Colon	19.54 ± 3.84 c	18.56 ± 1.87 b	-0.23	ns
Rectum	2.81 ± 0.27 d	6.49 ± 1.63 cd	2.41	*
Cecum	7.25 ± 0.67 b	1.66 ± 0.13 d	-8.21	***

*, **, and *** indicate significant interspecific difference (*p* < 0.05, *p* < 0.01, *p* < 0.001); ns indicates an insignificant difference. There are significant intraspecific differences indicated by the different lowercase letters (*p* < 0.05).

## Data Availability

The datasets used and/or analyzed during the current study are available from the corresponding author upon reasonable request.

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
