# Peer review of "A Comparative Study on the Distribution Pattern of Endocrine Cells in the Gastrointestinal Tract of Two Small Alpine Mammals, Plateau Zokor (Eospalax baileyi) and Plateau Pika (Ochotona curzoniae)"

_animals, 2023, doi:10.3390/ani13040640_

Round 1

Reviewer 1 Report

Manuscript entitled “A Comparative Study on the Distribution Pattern of Endocrine Cells in the Gastrointestinal Tract of Two Small Alpine Mammals, Plateau Zokor (Myospalax baileyi) and Plateau Pika (Ochotona curzoniae)” for Animals.

 The authors compared the distribution pattern of argyrophilic cells and the expression of 5-HT cells, Gas cells, Glu cells in the GIT of the plateau zokor and plateau pika by using histochemical and immunohistochemical methods. The results showed that the distribution pattern of endocrine cells in the GIT of plateau zokor and plateau pika not only have specific interspecies differences, but also exhibit parallel adaptation to the alpine environments.

The paper is clear and well detailed. The purpose of the study is appropriately defined. The details of the methods are comprehensible and consistent. The results are clearly presented. The discussion is relevant and complete. The manuscript contains sufficient and proper references. The Table is correct. The figures are good. Therefore I would suggest accepting the manuscript for publication in Animals after minor revision.

a)                  It would be interesting that the authors present some photos with H-E in order to accurately identify the morphology of the analyzed organs of the GIT.

b)                  The cells in the following figures are not clear enough and should be replaced

Figure 4. J, O, P, Q, S

Figure 6. C, E, S

Figure 8. A, F, M, P, Q

Reviewer 2 Report

                         Comments and Suggestions for Authors:

This manuscript focuses on the difference of the distribution pattern of endocrine cells in gastrointestinal tract between plateau zokor and plateau pika, and preliminarily explored the adaptation strategies of endocrine cells in the GIT of two small alpine mammals to the different foods in alpine environments. The paper has clear logic, appropriate workload, and reasonable literature citation, and the results have important reference value for further understanding the biological characteristics and ecological differences of these two species. In general, it has reached the quality requirements for publication. However, after reading this manuscript carefully, the article needs to be further improved by the following comments.

1. Line 4: The Latin name of plateau zokor in the title should be changed from Myospalax baileyi to Eospalax baileyi.

2. Line 101-103: The distribution pattern of endocrine cells includes the positive expression of various endocrine cells, so “and expression” should be deleted here.

3. Line 113-114: How to identify the captured animal as an adult? please explain here.

4. Figure 1: Because the latitude and longitude coordinates already exist in the map, the compass should be deleted according to the map specification.

5. Line 173-176: How to calculate the proportion of the micrograph image area that was positively stained? please elaborate.

6. Figure 2: It is recommended to mark different shapes of argyrophilic cells with different color arrows.

7. The position of the scale in all micrographs should be adjusted uniformly, such as Figure 2F, Figure 4N.

8. Line 249-252: The morphological description of these three endocrine cells after staining should be supplemented here, because the morphology of endocrine cells can be seen from the microscopic images.

9. Line 278-279: It is suggested that the title “The difference in the proportion of the micrograph image area positively stained for 5-HT-immunoreactive cells in the Gastrointestinal organs of the plateau zokor and plateau pika” of Figure 5 be modified to “The difference in positive expression levels for 5-HT-immunoreactive cells in the Gastrointestinal organs of the plateau zokor and plateau pika”.

10. Line 284-285: It is suggested that the title “The difference in the proportion of the micrograph image area positively stained for Gas-immunoreactive cells in the gastrointestinal organs of the plateau zokor and plateau pika” of Figure 7 be modified to “The difference in positive expression levels for Gas-immunoreactive cells in the gastrointestinal organs of the plateau zokor and plateau pika”.

11. Line 296-297: It is suggested that the title “The difference in the proportion of the micrograph image area positively stained for Glu-immunoreactive cells in the gastrointestinal organs of the plateau zokor and plateau pika” of Figure 9 be modified to “The difference in positive expression levels for Glu-immunoreactive cells in the gastrointestinal organs of the plateau zokor and plateau pika”. 

12. Line 377-380: What does the result of the difference in the proportion of opened-type argyrophilic cells between the two animals indicate? and what is the effect on the digestion strategy of the two animals? Please supplement the discussion.

13. Line 400-401: Is there any literature support? The references need to be added here.

Round 2

Reviewer 2 Report

The authors have solved my concern.